# BlotIt—Optimal alignment of Western blot and qPCR experiments

**Svenja Kemmer**[1,2]* , **Severin Bang**[1,2] , **Marcus Rosenblatt**[1,2] , **Jens Timmer**[1,2,3] , **Daniel Kaschek**[1,2,4]

**1** Institute of Physics, University of Freiburg, Freiburg im Breisgau, Germany, **2** FDM - Freiburg Center for Data Analysis and Modeling, University of Freiburg, Freiburg, Germany, **3** Signalling Research Centres BIOSS and CIBSS, University of Freiburg, Freiburg, Germany, **4** IntiQuan GmbH, Basel, Switzerland

☯ These authors contributed equally to this work.
* svenja.kemmer@fdm.uni-freiburg.de

**Data Availability Statement:** All relevant data are within the paper and its Supporting information files. The source code of the package is provided at https://github.com/JetiLab/blotIt.

## Abstract

Biological systems are frequently analyzed by means of mechanistic mathematical models. In order to infer model parameters and provide a useful model that can be employed for systems understanding and hypothesis testing, the model is often calibrated on quantitative, time-resolved data. To do so, it is typically important to compare experimental measurements over broad time ranges and various experimental conditions, e.g. perturbations of the biological system. However, most of the established experimental techniques such as Western blot, or quantitative real-time polymerase chain reaction only provide measurements on a relative scale, since different sample volumes, experimental adjustments or varying development times of a gel lead to systematic shifts in the data. In turn, the number of measurements corresponding to the same scale enabling comparability is limited. Here, we present a new flexible method to align measurement data that obeys different scaling factors and compare it to existing normalization approaches. We propose an alignment model to estimate these scaling factors and provide the possibility to adapt this model depending on the measurement technique of interest. In addition, an error model can be specified to adequately weight the different data points and obtain scaling-model based confidence intervals of the finally scaled data points. Our approach is applicable to all sorts of relative measurements and does not need a particular experimental condition that has been measured over all available scales. An implementation of the method is provided with the R package blotIt including refined ways of visualization.

## Introduction

The approach of mathematical modeling to analyse and understand dynamic processes of biological systems requires the collection and quantification of time-resolved experimental data for many different experimental conditions [1–4]. Frequently, the generation of these type of data is achieved by techniques like Western blotting [5, 6], quantitative real-time polymerase chain reaction [7], reverse phase protein arrays [8] or flow and mass cytometry [9] which only generate measurements on a relative scale. Therefore, the number of experiments that are

**Funding:** This work was supported by the German Ministry of Education and Research (BMBF) through the grants LiSyM (Grant No. 031L0048) and AML_PM (Grant No. 01KU1902A) as well as by the German Research Foundation (DFG) through the grants TRR 179 (Grant No. 272983813) and INST 35/1134-1 FUGG. The authors acknowledge support by the Open Access Publication Fund of the University of Freiburg and the state of Baden-Württemberg through bwHPC. The funders had no role in study design, data collection and analysis, decision to publish, or preparation of the manuscript.

**Competing interests:** The authors have declared that no competing interests exist.

comparable to each other, i.e. provided on the same measurement scale, is typically limited by the experimental setup, which constitutes a bottleneck for mathematical models with high complexity.

In the following we focus on Western blotting as a well-established and commonly used technique. In this technique, protein abundances are measured on a relative scale by chemi-luminescent antibodies binding to the respective proteins embedded in a gel. Let us consider such a time-course experiment that has been performed twice and quantified with Western blot. The experimental setting is assumed to be the same between the two experiments, i.e. the same biological or experimental conditions were measured, but on two different gels. Since the Western blot technique only provides a relative measurement, the obtained data points presumably show a similar dynamical behavior. However, they do not coincide with each other in absolute numbers due to experimental errors and different measurement scales. The corresponding unknown scaling factor, i.e. the ratio between the measurement scales, can be inferred relatively simply by aligning both measurement profiles to each other.

Now, let us consider another experiment where time courses of two different experimental conditions, as for example stimulation doses, have been measured separately on two different Western blots. Here, it cannot be distinguished whether the difference in the results is occurring due to the experimental condition or due to the different measurement scales. In particular, the scaling factor between the two blots cannot be estimated in this case.

One way to circumvent the missing comparability between measurements is to add recombinant proteins to the Western blot samples allowing for an absolute-scale quantification [10]. However, this approach is very expensive and time-consuming and scales with the number of measured proteins [11]. In systems biology, where the data is employed to estimate parameters of a mathematical model, it is a common approach to determine the scaling parameters of the different blots together with the remaining parameters of the model [12]. Besides the disadvantage of enlarging the parameter space when using standard ODE modeling and optimization methods, the estimates of the scaling parameters might be biased by the model equations, hampering hypothesis testing and therefore interpretation of the results [13].

As a generally applicable alternative, the Western blot experiments can be designed in a way that a certain experimental overlap exists between different blots, meaning that the same experimental condition is measured multiple times. Degasperi et al. [14] present a method to analytically determine the corresponding scaling factors based on such data. However, to be able to apply this method, there needs to be at least one experimental condition that has been measured on all blots which implies additional planing effort, might be limited by the availability of the overlap sample and complicates the use of experiments performed at a later time point.

Here, we present a new data-based approach for the estimation of scaling parameters which is also applicable in the absence of a unique condition overlapping across all available scales. It is sufficient when the independent experiments are connected by pairwise overlapping conditions. In addition, the implemented method provides not only the possibility to obtain scaling parameters and therefore align data points of different Western blots but also to compute confidence intervals for the results by applying a user-defined error model [15]. We implemented this method in the R package blotIt.

## Methods

When analyzing the measured values of a hypothetical experiment, we define three classes of effects: (A) *Biological effects* describe biological conditions as for example different targets

(proteins, mRNA, etc.), stimulation doses, inhibition treatments or measurement time points of a dynamical process. The set $I$ contains all $N_I$ unique combinations of biological effects. For each element of this set $i \in (1, \ldots, N_I)$, there exists one *true value* $y_i$. (B) *Scaling effects* describe the systematic influence of the measurement techniques and evaluation routines on the particular numerical value that is obtained. In the example of Western blotting, these scaling effects include for example development time, sample loading, gel thickness or antibody efficiency. All $N_J$ scaling effects make up the set $J$, and each scaling factor $s_j$ with $j \in (1, \ldots, N_J)$, equally affects the measurements of all $y_i$ within the respective experiment. Only measurements that underlie the same scaling factor can *a priori* be considered as comparable. In other words, repeated experiments measuring the same effect $y_i$ result in a set of values $Y_{ij}$, where the indices imply that $y_i$ is affected by experiment-specific scaling factors $s_j$. Some properties, e.g. gel imperfections, do not affect the whole experiment uniformly, but neighbouring lanes can be influenced by a systematic error. To resolve this, randomized sample loading is advised [16], ensuring that the resulting errors are independent. (C) *Residual noise*: In addition to the systematic error sources (A-B), each measurement $Y_{ij}$ is affected by stochastic noise $\epsilon_{ij}$. Based on these three error sources, we present in the following an approach to align the numerical values of different experiments with the aim of retrieving one comparable data set.

## Definition of the alignment model

In mathematical terms, the influence of scaling factors $s$ on true values $y$ is described by

$$Y = f(y, s) + \epsilon, \tag{1}$$

where $f(y, s)$ is the scaling model and $\epsilon$ reflects the noise of the measurement which is assumed to be normally distributed with $\epsilon_{ij} \sim N(0, \sigma_{ij}^2)$, where $\sigma_{ij}$ is the standard deviation of the normal distribution, and the indices imply that each measurement $Y_{ij}$ can in principle have its own error distribution. To assess the individual error distribution, an error model $h$ is introduced

$$\sigma_{ij} = h(y, s, e) \tag{2}$$

which is based on the error model parameters $e$. The data quantification happens usually by relating the luminescence of a sample to signal strength. An example for such a measurement procedure is Western blotting. Because of an always present background, it is in the nature of such measurements to have a low signal-to-noise ratio for data points with low signal. Kreutz et al. elaborate why the error of such measurements is most completely described by a mixed effects error model $h_{\mathrm{mixed}} = e_{\mathrm{abs}} + e_{\mathrm{rel}} \cdot f(y, s)$ composed of an absolute error addressing the constant background and a signal dependent relative error [15]. In cases where the signal is significantly higher then the background, or a constant background is subtracted in data quantification, it can be sufficient to describe the error by a purely relative error model $h_{\mathrm{rel}} = e_{\mathrm{rel}} \cdot f(y, s)$, although this simplified model still bears the danger to underestimate the errors of low intensity measurements, e.g. of unstimulated controls. In the following, we consider this relative error model. Kreutz et al. suggested statistical tests to check if this simplification is justified for a given data set [15].

Calculation of the errors by use of an error model has an additional advantage over the calculation by replicate spread. The error model considers the variance information of all experimental data, what allows for a reliable error estimate even for conditions with small numbers of replicates.

When considering Western blot data as a typical use case for the here presented method, a simple model with gel-dependent scaling effects is usually assumed. The equation then reads

$$Y_{ij} = f(y_i, s_j) + \epsilon_{ij} = \frac{y_i}{s_j} + \epsilon_{ij}, \tag{3}$$

where the measurement $Y_{ij}$ corresponds to the true value $y_i$ affected by the scaling factor $s_j$ and the noise $\epsilon_{ij}$. The relative error model for the standard deviation for the measurement $Y_{ij}$ then reads

$$\sigma_{ij} = h(y_i, s_j, e_{\mathrm{rel}}) = e_{\mathrm{rel}} \cdot f(y_i, s_j) = e_{\mathrm{rel}} \cdot \frac{y_i}{s_j}. \tag{4}$$

One error parameter $e_{\mathrm{rel}}$ is determined for all measurements, from which the measurement errors $\sigma_{ij}$ are inferred by multiplying $e_{\mathrm{rel}}$ with the corresponding model evaluation. The accuracy of the estimated errors crucially depends on the validity of the chosen error model. Therefore, the error model needs to be adjusted for other applications.

Depending on the measurement technique, the data could be given on the logarithmic scale, and the error model $h$ has to be adjusted accordingly. This is the case e.g. for qPCR data, which is typically provided on $\log_2$ scale:

$$Y_{ij} = f_{\log_2}(y_i, s_j) = \log_2\left[f\left(y_i, s_j\right)\right] = \log_2\left(\frac{y_i}{s_j}\right) = \underbrace{\log_2(y_i)}_{y'_i} - \underbrace{\log_2(s_j)}_{s'_j} + \epsilon_{ij} \tag{5}$$

Here, the relative error model becomes an absolute one:

$$\sigma_{ij} = h_{\log_2}(y_i, s_j, e_{\mathrm{abs}}) = e_{\mathrm{abs}} \tag{6}$$

Together, Eqs (1) and (2) describe a combined scaling and error model formulation based on the assumption that true values of measurements are influenced by scaling factors and experimental noise. All parameters $y$, $s$ and $e$ are *a priori* unknown and have to be determined based on the data.

## From the original to a common data scale

Evaluating the data with the presented method results in three representations of the data, each of them with its own meaning and application in different contexts: (i) The *scaled* data representation contains the original replicate data transferred to the common scale, (ii) the *aligned* data representation reflects the underlying estimated true values, and (iii) the *predicted* value representation corresponds to model evaluations back on the original scale. The alignment process as described in the following is schematically visualized in Fig 1.

Initially, the numerical values of each experiment are on their own *original scale* shown by three simulated example time courses at the top of Fig 1. In particular, these measurements are not comparable to each other. Now, let us for the moment assume that an optimal set of parameters $(\hat{y}, \hat{s}, \hat{e})$ has been found. With these parameters, we define a *common scale* as the scale on which all measurements shall be directly comparable. The model assumes that all true values $y$ are on this common scale, and describes how the scaling has to be applied to a true value $y_i$ to match the respective measurement $Y_{ij}$. To retrieve the scaled values $Y_s$ from the respective measurements $Y$ under the scaling $\hat{s}$, the *inverse* of this model has to be evaluated

$$Y_s = f^{-1}(Y, \hat{s}) \tag{7}$$

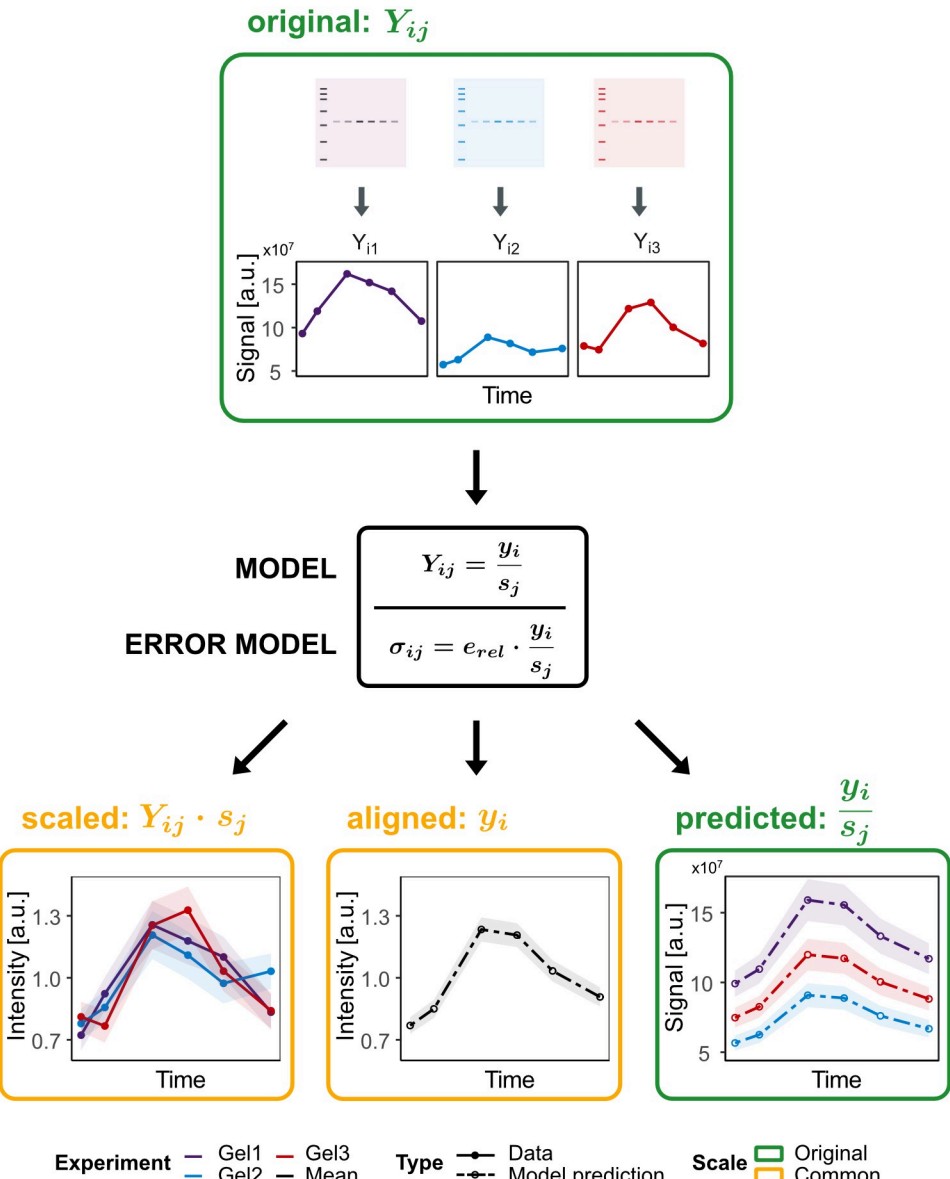

**Fig 1. Overview of the blotIt alignment procedure.** Top: Three exemplary experiments are represented by cartoon Western blots along with simulated raw data on the original scale (original). Experiments are indicated by color. Middle: Raw data is fitted by the alignment model to estimate scaling parameters $s_j$ and the underlying true values $y_i$. Error parameters $e_{rel}$ are simultaneously estimated by means of an error model. Bottom: The procedure outputs three different ways to visualize the result: Single replicates aligned to the common scale (scaled), the time course of estimated true values (aligned), and a prediction for the replicates on the original scale (predicted). Uncertainties are shown as shaded areas.

with estimated scaling parameters $\hat{s}$ and measured data $Y$ on the original scale. The resulting data set with replicates aligned to the common scale is thus called *scaled*, as shown on the lower left in Fig 1. Errors are not shown in Eq (7) because the model describes the scaling of the measured value itself. Error estimates for the measured data are derived from the error model and are propagated to the original scale by the use of Gaussian error propagation (see section *Error determination* for more details). The scaled data set still contains the information

about each independent experiment, but all measurement values are directly comparable. This is useful for the comparison between experiments, e.g. to determine potential outliers and identifying experiments with obviously significant measurement errors. As input for a dynamical modeling approach, the scaled data set is preferred in comparison to the original data as it is already on a common scale. Working with this data set, experiment-specific scaling parameters are not necessary anymore. To be able to compare this data on the common scale to model simulations on a different scale, e.g. absolute concentrations, it is recommended to include one scaling parameter for the whole data set in the model formulation. This enables a proper relation of model and data.

If there are only few replicates available, the scaled data set might not be the best input for dynamic modeling and the *aligned* data set should be favored. This data set consists of the estimated true values $\hat{\boldsymbol{y}}$. Corresponding estimated errors $\hat{\boldsymbol{e}}$ quantify the uncertainty of the parameter fit, thereby taking all data into account. These estimated errors might be a more reliable description of the data spread compared to the information provided by a small number of replicates.

To identify discrepancies between the true values determined based on all measurements, and the measurements of a single experiment, the true values can be scaled back to the original scale. This new data set is termed *predicted* and consists of the direct alignment model evaluations using the estimated true values and scaling parameters:

$$\boldsymbol{y}_{\mathrm{p}} = f(\hat{\boldsymbol{y}}, \hat{\boldsymbol{s}}) \tag{8}$$

Note that this data set is again on the original scale, and thus does not provide comparability between the experiments. The calculation of errors for the original, predicted and scaled data is described in the section *Error determination*.

## Parameter estimation

The presented method brings measurements from different experiments to a common scale applying an alignment and an error model. Assuming the parameters obey Gaussian statistics, the best maximum-likelihood estimate $\hat{\boldsymbol{\theta}} = (\hat{\boldsymbol{y}}, \hat{\boldsymbol{s}}, \hat{\boldsymbol{e}})$ is the set of parameters, which minimizes the negative log-likelihood [17]:

$$\hat{\boldsymbol{\theta}} = \arg \min_{\boldsymbol{\theta}}[-2 \log L(\boldsymbol{\theta})]. \tag{9}$$

Here, the log-likelihood function $\log L(\boldsymbol{\theta})$ consists of three terms:

$$\begin{aligned} -2 \log L(\boldsymbol{\theta}) = \ & \sum_{ij}\left(\frac{f(\boldsymbol{y}, \boldsymbol{s}) - Y_{ij}}{h(\boldsymbol{y}, \boldsymbol{s}, \boldsymbol{e})}\right)^2 && \text{(least squares)} & (10a) \\ & + \sum_{ij}\log(h(\boldsymbol{y}, \boldsymbol{s}, \boldsymbol{e})^2) + \log(\pi) && \text{(variance parameters)} & (10b) \\ & + \left(\frac{1 - \bar{y}}{10^{-3}}\right)^2 && \text{(normalization constraint)} & (10c) \end{aligned}$$

The special form of the log-likelihood function presented in Eqs (10a) and (10b) is based on the assumption that observations are affected by normally distributed residual noise. The first term (10a) includes the weighted least squares, namely the difference between model prediction $f(\boldsymbol{y}, \boldsymbol{s})$ and corresponding measurements $Y_{ij}$. Residuals are weighted by the variance $\sigma_{ij}^2 = h(\boldsymbol{y}, \boldsymbol{s}, \boldsymbol{e})^2$. The second term (10b) accounts for the simultaneous optimization of the

error model $h(\boldsymbol{y}, \boldsymbol{s}, \boldsymbol{e})$ and thereby estimation of error parameters. While the first two terms ensure minimization of the spread between experiments, the third term (10c) forces the mean of the estimated true values $\bar{y}$ to be one during the optimization process and thereby introduces the common scale.

For computational reasons, the parameters are per default transferred to logarithmic scale prior to the estimation. This drastically improves numerical stability especially when the input data varies over multiple orders of magnitude. Estimated parameter values are subsequently transformed back and reported on the linear scale.

## Error determination

In Fig 1, we introduced different output data representations as result of the alignment model. As a major advantage of this formulation, a measure of uncertainty, i.e. a statistical error, can be determined for each of these data sets, as summarized in Table 1 and explained in the following.

First of all, the number of estimated parameters $n_P = |\boldsymbol{\theta}|$ in the alignment model is typically quite high compared to the number of data points $n_d$: $n_d\,n_P \sim 2$–3. Under such conditions, the maximum-likelihood estimation tends to underestimate the standard deviation in a sample. The effect is more apparent for small samples or, equivalently, when many parameters are estimated from few data points. We account for this bias by applying Bessel's correction [18] to scale the estimated sample standard deviation by a factor

$$\gamma = \sqrt{\frac{n_d}{n_d - n_p}} \qquad (11)$$

that is multiplied with the estimated standard deviation of the measured data within blotIt:

$$\boldsymbol{\sigma}^{(\gamma)} = \gamma\boldsymbol{\sigma} = \gamma h(\hat{\boldsymbol{\theta}}) \qquad (12)$$

The error model is evaluated on the scale of the original observations. To retrieve the error of the scaled data, Gaussian error propagation is employed [19]:

$$\boldsymbol{\sigma}_s^{(\gamma)} = \left| \frac{\mathrm{d}}{\mathrm{d}Y}[f^{-1}](\boldsymbol{Y}, \hat{\boldsymbol{s}}) \right| \boldsymbol{\sigma}^{(\gamma)}. \qquad (13)$$

In Eq 4 we introduced a relative error model for Western blotting. As described above, this error serves as estimate for the error of both data sets, original and predicted. To retrieve the error of the scaled data $\boldsymbol{\sigma}_s^{(\gamma)}$, we have to consider the scaling model defined in (3).

$$\sigma_{s,ij}^{(\gamma)} = \gamma s_j \cdot \sigma_{ij}^{(\gamma)} \qquad (14)$$

While all errors considered until now quantify the uncertainty of the measurement, the error of the estimated true values $\hat{\boldsymbol{y}}$ are calculated qualitatively differently. Since $\hat{\boldsymbol{y}}$ are model parameters, their errors have to be estimated by the model uncertainty itself. In maximum

**Table 1. Overview table of the different output data sets of blotIt.**

| Data set | Data | Error | Scale |
|---|---|---|---|
| Original | $\boldsymbol{Y}$ | $\boldsymbol{\sigma}^{(\gamma)} = \gamma h(\hat{\boldsymbol{\theta}})$ | Original |
| Predicted | $\boldsymbol{y}_p = f(\hat{\boldsymbol{y}}, \hat{\boldsymbol{s}})$ | $\boldsymbol{\sigma}^{(\gamma)} = \gamma h(\hat{\boldsymbol{\theta}})$ | Original |
| Scaled | $\boldsymbol{Y}_s = f^{-1}(\boldsymbol{Y}, \hat{\boldsymbol{s}})$ | $\boldsymbol{\sigma}_s^{(\gamma)} = |f^{-1}(\boldsymbol{Y}, \hat{\boldsymbol{s}})|\boldsymbol{\sigma}^{(\gamma)}$ | Common |
| Aligned | $\hat{\boldsymbol{y}}$ | $\boldsymbol{\sigma}_{\mathrm{fit}}^{(\gamma)}(\hat{y}_i) = \gamma\sqrt{C_{ii}(\boldsymbol{\theta})}$ | Common |

likelihood estimation, the uncertainty is reflected in the local curvature of the likelihood land-scape around the determined parameter value [20, 21]. The uncertainty of the $l$-th fitted parameter $\sigma_{\text{fit}}^{(\gamma)}(\hat{\theta}_l)$ is given by

$$\sigma_{\text{fit}}^{(\gamma)}(\hat{\theta}_l) = \gamma \sqrt{C_{ll}(\boldsymbol{\theta})} \qquad \text{with} \quad C = I^{-1} = H^{-1}, \tag{15}$$

where the local curvature is approximated by the square root of the $ll$-diagonal element of the covariance matrix $C$ given by the Fisher information matrix $I$, which itself is represented by the Hessian $H$ that is calculated during the optimization process. Note that the Bessel correction $\gamma$ is applied here, too.

To sum up, the blotIt approach provides four measures of uncertainty (Table 1). Uncertainty provided with the original data set is estimated based on the error model. This error is reflective of the between-replicate variability and, as such, is comparable to the standard deviation of a single measurement. Uncertainty provided with the predicted data set is the same as for the original data set and, thus, reflective of the standard deviation of the single measurement. The uncertainty provided with the scaled data set is the error of the original data set translated to a different scale, i.e. the common scale. Also this error is reflective of the standard deviation of the single measurement. Finally, uncertainty provided with the aligned data set is the estimation uncertainty of the estimated *true* concentrations, meaning that this error is comparable to the standard error of the mean. Therefore, with more and more replicate measurements, the error of the aligned data set becomes smaller, whereas the errors provided with original, predicted, and scaled data sets consolidate.

## Simulation study

To compare the performance of different scaling approaches, we generated a simulated data set following a function with quadratic rise and exponential decay that represents the typical behavior of e.g. protein phosphorylation or expression dynamics [15].

$$f(t) = 0.1 + \frac{c_{\text{cond}} \cdot 10^{-3} \cdot t^2}{\exp\left[\dfrac{t}{50 \cdot c_{\text{target}}}\right]} \tag{16}$$

The parameters $c_{\text{cond}}$ and $c_{\text{target}}$ were chosen from a uniform distribution $\mathcal{U}(0.5, 1.5)$ for each condition and target to simulate different stimuli and target specific dynamics. The fixed values were just used to determine the time scale of the dynamics.

An artificial noise consisting of an absolute contribution resembling background noise, as well as a signal dependent relative part, was added to the simulated data:

$$f_{\text{noise}} = f(t) \cdot \epsilon_{\text{rel}} + \epsilon_{\text{abs}}. \tag{17}$$

Because this noise is known to be log-normally distributed, the error was implemented as

$$f_{\text{noise}} = \exp\{\log[f(t)] + \mathcal{N}(0, \sigma_{\text{rel}})\} + \exp[\mathcal{N}(0, \sigma_{\text{abs}})], \tag{18}$$

where $\mathcal{N}(0, \sigma)$ describes a Gaussian distribution with mean 0 and standard deviation $\sigma$.

To evaluate the goodness of the scaling and compare blotIt with other available methods, the same generated noisy data was scaled with different normalization approaches. This procedure was repeated 200 times for each normalization approach to be able to statistically evaluate the results. Inspired by Degasperi et al. [14], the goodness of each normalization was assessed based on the spread of the scaled data, the standard deviation *sd*. It was calculated for each

biological condition *i*, determined by target, condition, and time point:

$$\sigma^i_{\text{calc}} = sd\left\{\log\left[\frac{\mathbf{Y}^i_{\text{s}}}{\overline{\mathbf{Y}}^i_{\text{s}}}\right]\right\}.\tag{19}$$

Since the data was generated with log-normally distributed noise, the data points had to be log-transformed before the standard deviation was calculated. The so calculated $\sigma^i_{\text{calc}}$ represents the normalized spread of the scaled replicates for each biological condition and is comparable between scaling methods.

## Applications

In the following, the application of blotIt is illustrated and compared to alternative approaches by means of simulated data and a published data set comprising Western blot as well as qPCR measurements. The scaling and thereby alignment of the different data sets was performed with the R package blotIt.

### Application to simulated data

To assess the performance of blotIt in comparison to alternative normalization approaches, we conducted a method comparison. Three alternative methods were tested on data realizations with different overlap, i.e. number of samples measured in all experiments, and different noise level. Their performance was compared to the one of blotIt. The following methods were analyzed: (1) Optimal alignment, which is based on the analytical minimization of differences between all overlap samples. This approach was discussed in detail by Degasperi et al. [14] and applied e.g. by Wang et al. [22]. (2) Normalization by fixed point, which uses one biological condition (one $y_i$ as defined above) to normalize all experiments by the respective measurement value of this condition [23, 24]. (3) Normalization by sum (setSums), or equivalently average, which is analog to the fixed point method, but here experiments are divided by the sum or average of all overlapping biological conditions [25, 26].

We compared the performance of the different scaling techniques for five data realizations chosen to mimic a variety of real world situations: *Full overlap*—Each experiment describes the exact same biological conditions, meaning the exact same experiment was repeated *N* times; *50% overlap*—In this scenario all experiments share one reference treatment condition, for which all time points are measured. The second half of each experiment covers an individual condition; *Dose response*—Here, a whole time course is measured in replicates analogously to the previous scenarios, along with an additional replicate set that covers just one time point in multiple conditions. This could be e.g. a dose response measurement; *Signal to noise variations*—Two more data sets with 50% overlap were simulated with mixed signal to noise ratios. All data sets are visualized in Fig 2a. Experiments describing the exact same biological conditions are referred to as replicate sets indicated by color.

The performance of the individual methods was evaluated for each of the data realizations based on the spread of the scaled data, the standard deviation $\sigma_{\text{calc}}$, as described in the methods section. As the performance might vary with the number of replicates, i.e. the number of experiments belonging to one replicate set, one scenario with three and one scenario with ten replicates was analyzed. As displayed in Fig 2b, all methods performed equally well in the scenario with full overlap except for the fixed point approach that had a slightly worse outcome. However, with decreasing number of overlap samples blotIt gained advantage over the other methods tending towards overall smaller standard deviations. The scaling evaluation of the 50% overlap data set was especially interesting as it visualized how the distinct approaches work. The three methods checked against blotIt displayed a small sharp peak for low standard

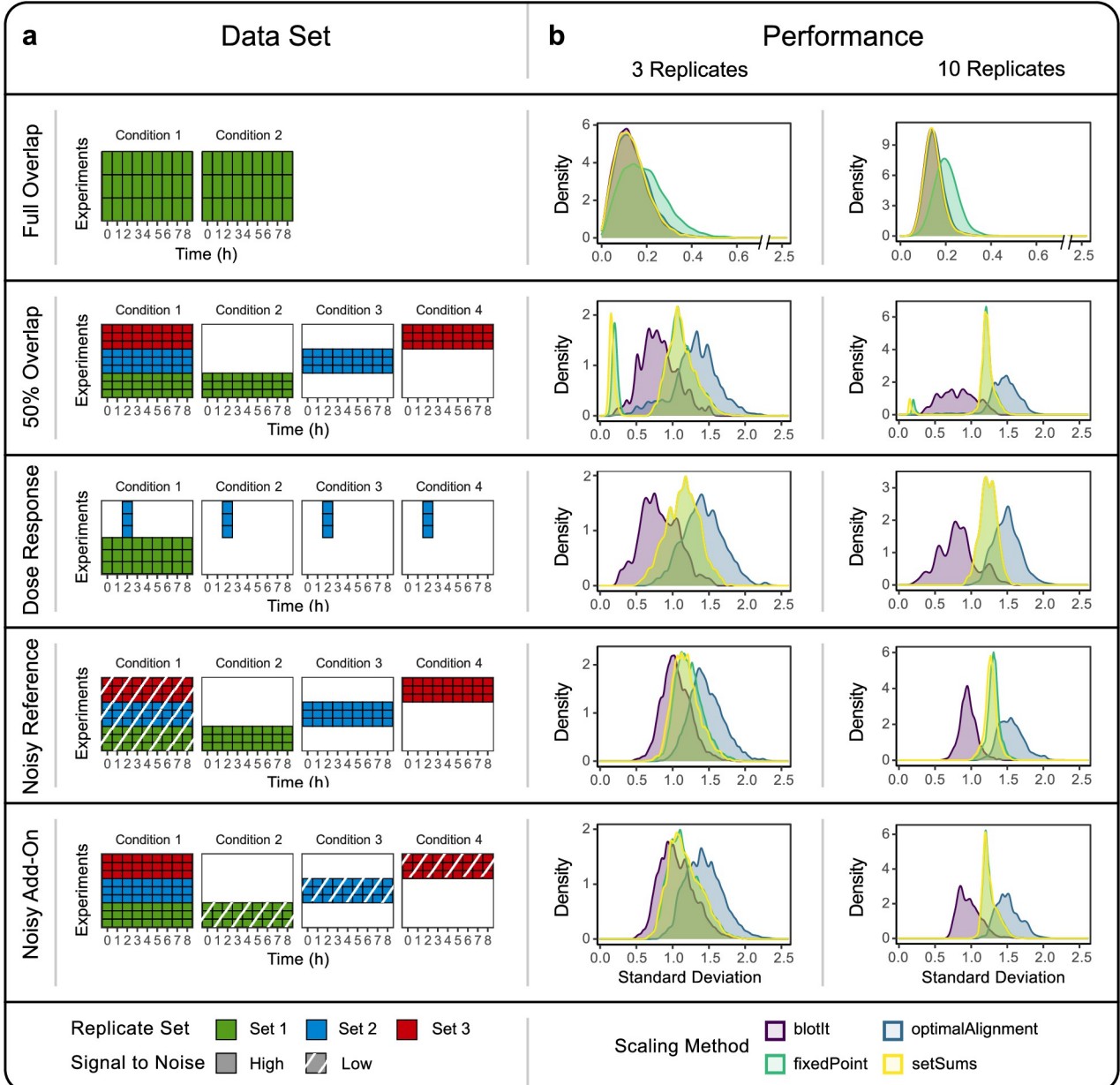

**Fig 2. Method comparison.** The performance of four different scaling methods was analyzed for five simulated data sets with different overlap and signal to noise ratios. (a) Illustration of the tested data sets and their experimental overlap. Rows of the tile plots correspond to the different experiments (scaling effects), columns correspond to different experimental conditions (biological effects). Tiles indicate whether the respective condition was measured in the respective experiment (colored) or not (white). Those experiments describing the exact same biological conditions are summarized and colored as replicate sets. Data with low signal to noise ratio is indicated by shaded area. (b) The performance of the different scaling methods was assessed based on the standard deviation of the respective scaled data and displayed as density plot. Data sets were analyzed with three replicates, i.e. replicate sets consisting of three experiments and with ten replicates, respectively. Note that the methods *setSums* and *fixedPoint* often yield very similar results and thus lead to overlaying density plots.

deviations comparable to results from the perfect overlap scenario, followed by a larger peak for higher sigmas. This characteristic was less marked in the optimal alignment approach. Except for blotIt, all methods just use the overlap present between all experiments to determine the scaling factors. The just described small peaks for low sigmas were originating exactly from

these overlap samples. Yet, conditions not measured in all experiments were scaled a lot worse and led to a larger spread between replicates. In contrast, blotIt uses all samples to normalize the data. This improves the scaling of samples not measured in all experiments and of data with low signal to noise ratio as present in data sets *NoisyRef* and *NoisyAddOn*. Frequently, a small overlap between experiments is encountered in the typical scenario of combining time course and dose response measurements. At the extreme, when the overlap between experiments was reduced to one condition, blotIt outperformed the other methods by far. With only one overlapping condition, normalization by fixed point and by sums were equivalent in this scenario. Optimal alignment relied on the common condition to calculate the scaling factor, which gave the worst outcome, indicating that this method works best when a large overlap is provided. It has to be noted, however, that the overall scaling performance for all methods got worse with less overlap.

Comparing the outcome in regard to replicate numbers, similar means of the standard deviations could be observed for three and ten replicates. The overall performance of the methods thus did not change. However, With increasing number of replicates the peaks got sharper. The equal performance might be due to the design of the data sets, where additional replicates were evenly distributed between reference and individual conditions and thus did not change the proportion of data used for scaling.

## Alignment of Western blot data

In the following, blotIt is applied to a published data set that provides time-resolved measurements for the phosphorylation of cytoplasmic *Signal Transducer and Activator of Transcription 1* (STAT1) and mRNA levels of the *Suppressor of Cytokine Signaling 1* (SOCS1) [4]. Both targets are involved in the Interferon alpha (IFNα) signaling pathway, where STAT1 acts as a transcription factor regulating, among others, SOCS1 expression. Three different IFNα concentrations were used to induce signal transduction and thereby phosphorylation of STAT1 as well as expression of SOCS1. Phosphorylation dynamics of cytoplasmic STAT1 were quantified by Western blot experiments, while SOCS1-mRNA levels were measured by qPCR and are analyzed in the next section. The three IFNα doses used for stimulation correspond to three conditions that are distinguished in the following.

Cytoplasmic STAT1 protein was quantified in three experiments. Measurements are thus only available on different scales as they originate from distinct gels. Therefore, one can not directly judge the dynamics of the final time course from investigating the raw data (Fig 3a). Moreover, a direct comparison for example between experiment two and three is not possible, since these have not been measured together on the same gel. Instead, as shown in Fig 3b, an overlap exists between experiments 1 & 2, and 1 & 3 by two replicates, respectively. This allows the alignment of the three experiments and enables a comparison between all replicates. If no overlap existed between gels, it would not be possible to determine or define a common scale. In this case, blotIt would determine scaling factors separately, and the resulting scaled values would not be comparable between experiments.

Within blotIt, scaling of these Western blot data is performed via the alignment function `alignReplicates()` called as

```
outputWB <- alignReplicates(
  data = mydata,
  model = "yi/sj",
  errorModel = "e_rel*value",
  biological = yi ~ name + time + condition,
  scaling = sj ~ name + gelID,
  error = e_rel ~ name)
```

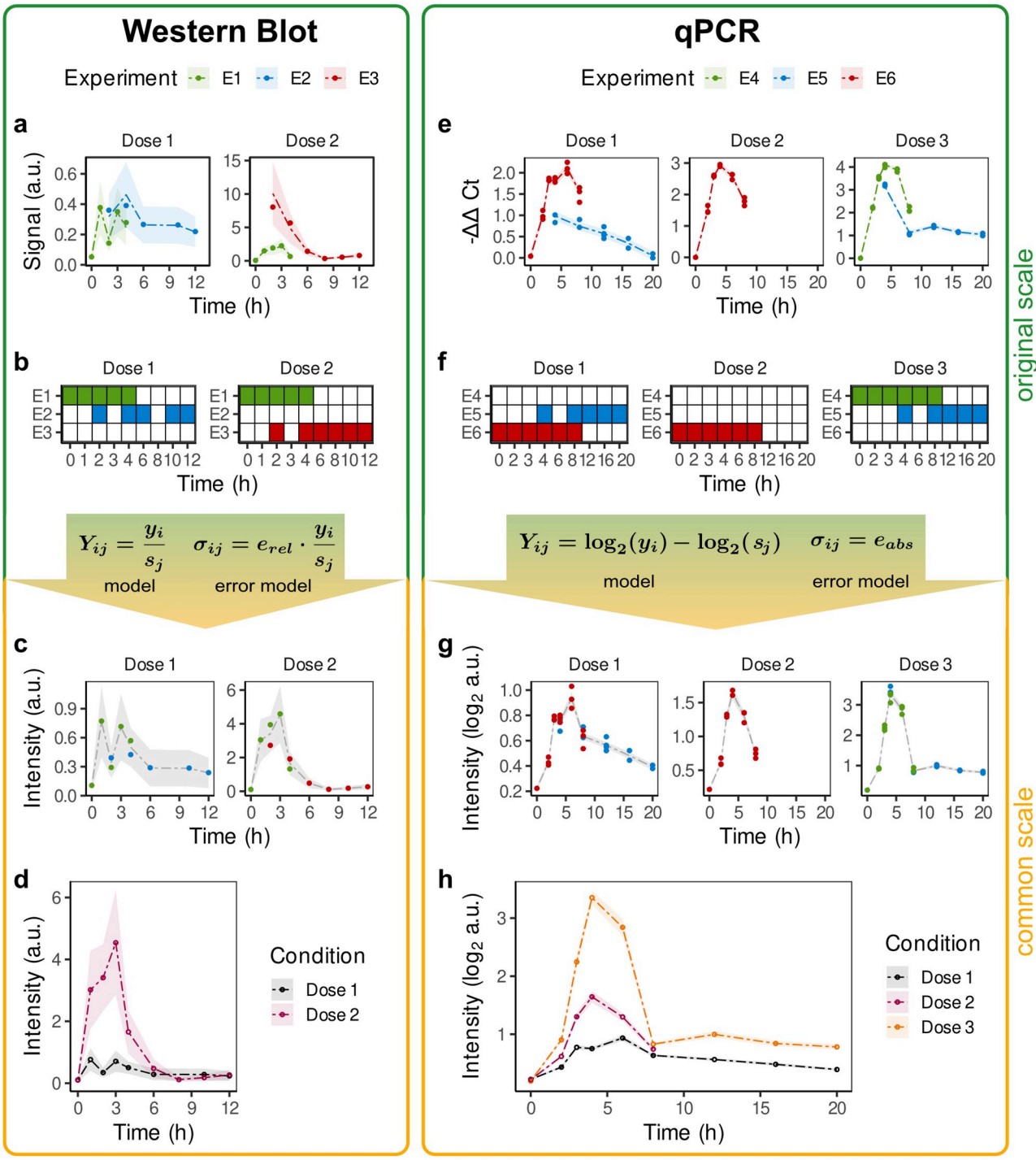

**Fig 3. Application example for the alignment of Western blot and qPCR data.** Raw data of cytoplasmic pSTAT1, measured by Western blot, and SOCS1 mRNA quantified with qPCR, was taken as a subset from [4]. (a, e) Raw data is shown on the original scale (dots) compared to the predictions (dashed interpolating lines) as output by the model. Color indicates the different experiments (gels). (b, f) Illustration of the experimental overlap. Rows correspond to the different gels (*scaling effects*), columns correspond to different experimental conditions (*biological effects*). Tiles indicate whether the respective conditions was measured on the respective gel (colored) or not (white). (c, g) Data points after the alignment are shown. Scaled replicates (dots) are colored according to their original gel. On the same common scale, estimated true values are shown as gray interpolating lines. (d, h) Aligned data (dots) and trajectories (linearly interpolating lines) are depicted on the common scale. Color indicates the experimental condition.

allowing an individualized structure of model and error model. The structure of the input data set is oriented towards a recently developed data sharing standard for dynamic modeling, in particular the measurement file of PEtab [27]. It has to be provided as a `data.frame` with obligatory columns `name`, `time` and `value`, specifying the observed target, the measurement time and the measurement value. Further columns characterize additional biological effects, which have to be distinguished, and scaling effects, e.g. the experimental `condition` or the `gelID` of the Western blot.

The alignment function allows to define these effects as a function of $y_i$ or $s_j$ in an additive manner. `name` and `time` are obligatory for the argument biological, as different targets at different time points are independent and have to be distinguished. The scaling argument requires the parameter `name`, as every observed target may scale differently. Also the error, here `e_rel`, can be specified. If the same value of `e_rel` should be assumed for all conditions, the error is only specified by the target, i.e. `name`.

Alignment of the pSTAT1 data brings the measured data points from different experiments to a common scale. The `output` of `alignReplicates()` is a list with the entries *scaled*, displaying the scaled replicates $Y_{ij}*s_j$ (Fig 3c) and *aligned*, containing the estimated true values $y_i$ (Fig 3d), both with uncertainties. Further listed elements describe the original data, data predictions based on the alignment model, and the respective estimated scaling parameters.

## Alignment of qPCR data

The flexibility of the model based alignment approach allows to process qualitatively different data with only minor adjustments. Thus, in addition to the linear Western blot data, also quantitative real-time polymerase chain reaction (qPCR) data can be analyzed with blotIt. During the process of mRNA quantification in qPCR measurements, a small region of the mRNA of interest is amplified in a sequence of replication cycles. The mRNA concentration is therefore measured in Cycles to Threshold ($C_t$) of PCR, a relative value that represents the cycle number at which the amount of amplified DNA reaches a defined threshold level. This threshold is in general individually chosen for each experiment. Since the amount of mRNA is approximately duplicated in each cycle of the PCR, the $C_t$ value is on the log2 scale. The inferred quantity $\Delta C_t$ describes the difference in cycles between the target and a reference gene, where the reference gene can be a *housekeeper* which is known to remain relatively stable in response to any treatment. To assess the dynamic development of mRNA expression the $\Delta\Delta C_t$ can be used, representing the difference in $\Delta C_t$ between the target and a reference condition [28]. Here, an intuitive reference condition is the zero time point. Because higher $C_t$ values correspond to lower mRNA abundance in the sample, the quantity $-\Delta\Delta C_t$ is used for the description of expression dynamics.

The freedom of choice for the detection threshold results in an experiment-specific shift in the number of cycles until detection. Together with the offsets introduced in the $\Delta\Delta C_t$ calculation by potential measurement variances of the zero time point and the housekeeper genes, this can be summarized into one offset parameter. Since the data is of logarithmic nature, this offset reflects a multiplicative scaling on the linear scale. The alignment model and the corresponding error model thus have to be log-transformed, which results in an additive model with an absolute error description. The `alignReplicates()` function call is adjusted accordingly:

```
outputQPCR <- alignReplicates(
  data = mylog2data,
  model = "log2(yi)-log2(sj)",
  errorModel = "e_abs",
  biological = yi ~ name + time + condition,
```

```
scaling = sj ~ name + gelID,
error = e_abs ~ name)
```

The results of the alignment process `outputQPCR` are analogous to those described for the Western blot data. They are described for the example data set of SOCS1-mRNA on the right hand side of Fig 3e–3h). Data was quantified in three experiments, where experiment four and six are not comparable on the original scale and differences in conditions only become apparent on the common scale. Furthermore, the dynamics of the full time course are not visible at the level of the single experiments (Fig 3e) but can be analyzed after scaling in Fig 3h. Since the original scale is logarithmic this is also true for the common scale.

## Discussion

In many cases, biological data is generated in a way that does not allow a direct comparison between different measurements. Reasons can be differences in sample loading, antibody binding or discrepancies between various gels in the case of Western blotting. In turn, these artifacts lead to different measurement scales for the experimental data and mask the effects of biologically different conditions like treatments and measurement time points in dynamical processes.

Analyzing longitudinal data or dose response data without proper preprocessing is not possible when the measurements are affected by different scaling factors. We here present a method to scale data of independent experiments to one common scale, where the data is directly comparable. In addition to the *original* and the *scaled* data, we provide two further outputs of the algorithm: (1) *aligned* data, i.e. the *true values* obtained when the impact of different scaling and residual noise is removed; and (2) *predicted* data, i.e. the values on the original scale of the experiments obtained when only the impact of residual noise is removed.

Previously established strategies to correct for the scaling differences include the usage of recombinant proteins to transform each measurement to an absolute scale, or theoretical approaches like normalization by fixed point, by sum and optimal alignment. In the latter, scaling factors are determined by analytically minimizing differences in scaling between the experiments. The here presented method follows a similar idea but determines the scaling factors via a more flexible numerical optimization. Our approach has the benefit that no single condition needs to be present within all experiments, but instead it is sufficient to have a pairwise overlap of measured conditions between different experiments. Even if a certain overlap is given between all experiments, it is common that some conditions are measured only in a subset of experiments. In contrast to blotIt, the above mentioned approaches cannot use the data points outside of the overlap to determine the scaling factors, which are therefore scaled with poor quality. Here the power of blotIt comes in, taking all data into account for the estimation of the scaling factors. This is especially relevant when performing experiments for a lot of different conditions e.g. times or doses of stimulus or inhibitor. Further, in contrast to the other methods, we are not only able to determine the scaling factors, but also estimate underlying true values, i.e. maximum-likelihood estimates for the true values disregarding the experimental scaling artifacts. Asymptotic confidence intervals based on the Fisher Information Matrix are provided for the scaling factors as well as for the estimated true values.

One typical field of application for biological time course data is dynamical modeling where it is often beneficial to reduce the number of estimated parameters to a minimum, e.g. via data pre-processing. By determining a common scale with the presented method, it is not necessary to include experiment-specific scaling factors in the model, decreasing the parameter space and therefore the model complexity. A special optimization approach termed hierarchical optimizing also enables the calculation of scaling parameters without effectively increasing

the parameter space [29, 30]. However, this analytical evaluation of scaling parameters is always combined with the parameterization of an ODE model as it is the case for Weber et al. [13]. BlotIt as well as the other analyzed methods are model-free purely data-based approaches, i.e. methods not depending on a specific ODE model implementation or modeling framework. However, it might be of interest to investigate these integrative approaches in comparison to blotIt in a future study.

By utilizing numerical optimization, alignment model and error model can be flexibly adapted to the appropriate scaling mechanism for the data at hand. One can therefore account e.g. for data on the logarithmic scale or apply customized scaling approaches. The same freedom applies to the error model with the benefit to individually include relative and absolute errors. With this flexibility, blotIt can not only be applied to data generated by Western blotting, but to all use cases where relative data is generated like quantitative real-time PCR, reverse phase protein arrays as well as flow and mass cytometry.

## Supporting information

**S1 Raw images.**
(PDF)

## Acknowledgments

We thank Marcel Schilling and Elisa Holstein for instructive feedback on the experimental procedure of qPCR measurements.

## Author Contributions

**Conceptualization:** Daniel Kaschek.

**Funding acquisition:** Jens Timmer.

**Investigation:** Daniel Kaschek.

**Methodology:** Daniel Kaschek.

**Software:** Svenja Kemmer, Severin Bang, Daniel Kaschek.

**Supervision:** Jens Timmer.

**Visualization:** Svenja Kemmer.

**Writing – original draft:** Svenja Kemmer, Severin Bang, Marcus Rosenblatt.

**Writing – review & editing:** Jens Timmer, Daniel Kaschek.

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
