## [Decision Letter · Decision Letter 0]

2 Mar 2022

PONE-D-22-03759BlotIt - Optimal alignment of western blot and qPCR experimentsPLOS ONE

Dear Dr. Kemmer,

Thank you for submitting your manuscript to PLOS ONE. After careful consideration, we feel that it has merit but does not fully meet PLOS ONE’s publication criteria as it currently stands. Therefore, we invite you to submit a revised version of the manuscript that addresses the points raised during the review process.

 The three reviewers agree that the manuscript would be a valuable addition to the literature, provided that a number of aspects are clarified, and I concur with their assessment. In the review reports you will find a list of suggestions to clarify a number of technical aspects of the proposed method. Notably, two of them agree that it would be interesting to compare BlotIt with alternative approaches. As pointed out by Reviewer 2, one possibility would be to perform a comparison using simulated datasets. If possible, such a study would greatly enhance the contributions of the paper. 

We look forward to receiving your revised manuscript.

Kind regards,

Alejandro Fernández Villaverde, Ph.D.

Academic Editor

PLOS ONE

Journal Requirements:

Reviewers' comments:

Reviewer's Responses to Questions

**Comments to the Author**

1. Is the manuscript technically sound, and do the data support the conclusions?

Reviewer #1: Yes

Reviewer #2: Yes

Reviewer #3: Yes

2. Has the statistical analysis been performed appropriately and rigorously? 

Reviewer #1: Yes

Reviewer #2: Yes

Reviewer #3: Yes

3. Have the authors made all data underlying the findings in their manuscript fully available?

Reviewer #1: Yes

Reviewer #2: Yes

Reviewer #3: Yes

4. Is the manuscript presented in an intelligible fashion and written in standard English?

Reviewer #1: Yes

Reviewer #2: Yes

Reviewer #3: Yes

5. Review Comments to the Author

Reviewer #1: BlotIt - Optimal alignment of western blot and qPCR experiments

Summary

The authors present an automated procedure for the normalisation of relative data, such as data obtained from Western blot or rt-qPCR. Relative data are usually not directly comparable across replicates, because of different arbitrary units obtained, and so require normalisation. Using likelihood-based optimisation, the proposed method is able to estimate how to scale each replicate, and at the same time provides an estimate of mean and variance for the underlying data.

The approach is flexible in various way, as it allows different error models to be used, an also it is able to combine data from multiple blots as long as there is one experiment shared across pairs of blots.

The work presented is certainly worthy of publication, although some clarifications and minor corrections are necessary before I can give my final approval.

Major

1. Missing discussion of the fact that Yij at extremes of the dynamic range of detection can have relatively low signal to noise ratio or high residual error epsilon. While the error epsilon is initially introduced as potentially different for each Yij, it is then simplified to have a variance proportional to the value of the measurement (sigma_ij=e_rel*yi/sj). While this can be considered an appropriate generic error model, its drawbacks should also be highlighted, such as its underestimation of variance for relatively low intensity blot values (which notoriously have a low signal to noise ratio). It would also be more appropriate to adjust lines 80-83 page 3, to reflect the fact that this error model (Eq 2) efficacy depends on the assumptions chosen to implement h and whether the data follows these assumptions, rather than just describing it as a superior approach. The advantages and disadvantages of the chosen simplified model should be highlighted. I guess the advantage here is the reduction of parameters to estimate (just e_rel), while the disadvantage is the loss of model flexibility. For example, it will ignore that low intensity values might have a much higher relative variability (because of the low signal to noise ratio).

2. Please define a set or range of values for the indexes i and j. For example there could be I conditions and J blots with i in the range 1,…,I and j in the range 1,…,J. In principle, each equation should have a definition for i and j range, like ‘for all i in (…), j in (…)’. The simplest case would be when the set of experiments i is the same for all J. However, the authors imply that different j can have different sets of experiments, so different i? In this case perhaps it makes more sense to talk about indexes i that belong to sets Ij (that is indexed by j) and that the intersections of sets Ij need to be at least pairwise not empty (i.e. share at least one condition i). This might also affect the notation in equation 10.

3. How is the mean of the true values y dash (equation 10c) defined? This is particularly important if there are different conditions i on different blots j (like in Figure 2), I guess this will be a mean across all i regardless of what blot j they belong to?

4. In equation 7, page 4, the measurement error term is not shown. How do you expect the value of Ysij be determined if the error is unknown? For example, arranging Eq 1, Ysij = (Yij+eij)*(s hat j). Because the error is potentially different for each data point, how will the mean and sd of the aligned data be affected? More explanation is given later, but here a clarification and showing the particular example (f=Y/s) next the generic equation would help the flow. Perhaps, it would help to mention the ideas of error propagation here, with examples and simplifying assumptions.

5. At lines 129 to 131 page 4, the authors assert that for dynamic models parameter estimation mean and standard deviation is sometimes preferred, citing the case of low number of replicates. One could argue the opposite, that if there are few replicates, then all data should be used for fitting, because the average and sd alone may not properly represent a given datapoint. I suggest to remove or add more literature or explanation to support the authors claim.

6. The claim at lines 133 and 134 that the estimated errors are more reliable than the data spread obtained from replicates, should be further explained. For example, this could be true only if the data agree with the error models and there could be exceptions (see above for low signal to noise datapoints), and also it depends on how the data spread of replicates is calculated, for example if the replicates are done all on the same blot, then it might be more accurate, but if they are done on different blots, then the spread is completely dependent on the normalisation applied and possibly the value of other datapoints.

7. Why there is no error term in equation 8? Also, in this case it would be useful to give a generic idea of how the error can be inferred and follow Eq 8 with the concrete example f=Y/s, and simplifying assumptions.

8. On page 5 line 146, please reconsider and rephrase the claim that the optimal theta is obtained by minimising the spread of the replicates, because the likelihood model is not just about trying to reduce the spread (low sigma).

9. In the error determination section (page 6), equations 13 and 14 require a reference, please add. Also, it would be nice to accompany these equations with concrete examples for the specific cases and simplifying assumptions described (like f=Y/s). For example, sigma_s, if I am not mistaken, simplifies to sigma_s_ij=s_j*sigma_ij.

10. Perhaps it would be of interest adding to the discussion some speculative yet useful cases. For example, what happens when two blots share only one experiment, but in one blot j the measurement is likely to have a high signal to noise ratio, while in the other blot j’ the same experiment has a very poor signal to noise, perhaps because it is a low intensity value. Would the proposed model be able to propagate the variance of the normalised data? Would there be enough data to constrain the model optimisation?

11. What other error models are available in blotIt besides e_rel*value and e_abs?

Minor:

12. Shouldn’t Equations 5 and 6 mirror the definitions in Equations 3 and 4? Equations 5 and 6 seem to be a mix of the generic vector format Equations 1 and 2 and equations with specific indexes i and j such as Eq 3 and 4. Please, choose one format for clarity, probably the format with the indexes i and j would be more suitable for Eq 5 and 6 following the flow of the paper.

13. Line 114, page 4, what is the dimensionality of y hat, s hat and e hat? Probably this will be clearer once the indexes are clarified (see above)

14. ‘Therefore’ might be more appropriate than ‘therefor’ at line 8 page 1, and also line 50 page 2, and also in other places across the manuscript.

15. Line 118 page 4, form -> from

16. In equation 7 page 4, the variable Y_s is undefined. I was initially confused by it because the text that precedes it talks about the true values y. It might help to write a sentence giving a proper definition for Y_s, and accompany this equation with another equation exemplifying what Y_s_ij look like for the simplified model f=Y/s and error e_rel*Yij/sj.

Reviewer #2: This manuscript describes a normalization strategy for western blotting as well as many other assay types that can put relative data from different experiments or replicates on the same quantitative scale. This is needed often because experiment specific factors cause the scaling to not be comparable between replicates or different experiments. A main claimed novelty is that the same condition need not be contained in every experiment for normalization. Rather, each experiment needs to share at least one point with one other experiment. I would have liked to have seen more application and discussion to data sets that have such a feature, and showing where current methods fail. And perhaps some discussion of how often this scenario is found in the literature and would be needed.

There are other points listed below that may be important to address:

1. What justification do the authors have for assuming normality in errors for western blots? How much does that affect the conclusions of the paper? Could alternative error models be used and BlotIt still functions well? How does that impact application to other data types?

2. It is appreciated that the authors technique is claimed to work when data points are not shared between all replicates and/or conditions. How common or rare this is for bench scientists performing replicates or experiments was not discussed. What novelties or advantages does blotit have when data points are shared among all replicates or experiments? Getting more clarity on that would help make the impact and uptake of the paper clearer.

3. How does the proposed approach differ from the scaling factor approach described here: 10.1038/msb.2009.4 ? There is a general lack of comparison to other analysis methods which have been established and used for quite a long time, as cited by the authors.

4. The discussion of how to use the R code and format it seems more appropriate for detailed methods section, not the results section.

5. In Figure 2, how do alternative methods for normalizing data compare to BlotIt?

6. Perhaps a simulated data study where data points are actually shared between all experiments, but are hidden to see how BlotIt does, could be an effective analysis to demonstrate usefulness and also compare to other normalization methods.

7. How would one compare the common scale data as the output of BlotIt to model simulations that would have a different scale (e.g. absolute concentrations)? Often comparison to and use with dynamical models is cited in the paper as a main motivator, but discussion with respect to this is lacking.

8. Therefor therefore

Reviewer #3: Review for the manuscript "BlotIt - Optimal alignment of western blot and qPCR experiments":

The manuscript proposes a novel alignment method for relative data. Via optimization, it finds a version of the data on a common scale. An implementation in R is provided.

The manuscript is overall well written and easy to read, while in some places it could be more specific. In my opinion, the new method is interesting and will find usage, while it is maybe not a mayor conceptual breakthrough.

I have a few comments, which I think should be addressed in a revision:

Content

-------

- it could be made clearer that, unlike e.g. the approach by Weber et al., the approach is essentially model-free, i.e. only dependent on the data and a noise model, but not e.g. a post-hoc employed ODE model.

- A comparison to e.g. the method by Weber, as well as the method by Degasperi, in a situation where it is applicable, would be of interest, e.g. regarding predictions, efficiency and uncertainties. It is however understandable if this is beyond the scope of this work.

- l. 35f: "disadvantage of enlarging the parameter space drastically": The approach by Weber (or also later papers by Loos et al. "Hierarchical optimization for the efficient parametrization of ODE models" and Schmiester et al. "Efficient parameterization of large-scale dynamic models based on relative measurements") appear to argue explicitly that the parameter space is effectively not enlarged by a hierarchical formulation.

- l. 36f: I did not understand "estimates of the scaling parameters might be biased by the model equations[,] hampering hypothesis testing and therefor[e] interpretation".

- l. 82: "the error model considers the variance information from all experimental data" as opposed to "calculation of errors based on the spread of measurement values" does not get clear to me.

- l. 146: There appears to be a constant $\\pi$ missing in (10b), when deriving (10a) and (10b) from a normal density, which affects the relative impact of both terms.

- l. 146: I think it could be clarified that $\\bar y$ in (10c) denotes the mean (over all data points?)?

- l. 146: Where does the $10^{-3}$ come from? This appears to be rather arbitrary and may affect how much emphasis the method puts on normalization. How sensitive is the method with regard to it? Or can it be chosen in a problem-specific manner? Or would there be an alternative formulation as an optimization problem with explicit constraint $\\bar y = 1$?

- l. 146: Can (10c) be interpreted stochastically (as (9) claims to describe a probability density)?

- l. 158: "This drastically improves numerical stability": This is surely an accurate fact, yet a reference may be good. Is the method applicable to negative data?

- l. 167ff: References on the parameter/data ratio, the variance underestimation, and the Bessel correction would be good.

- l. 180: The confidence interval appears based on a local Taylor approximation given asymptotic normality of the maximum likelihood estimate (with covariance matrix given by the inverse Fisher information matrix). Conceptually, there should be alternative methods, e.g. based on Wilk's theorem or sampling. Maybe a contextualization would be good?

- l. 184: What do the authors mean by the FIM is "represented by the Hessian"?

- implementation of the method: How is the optimization problem solved? Are gradients available? Does the problem have multiple local optima?

- implementation of the method: How computationally expensive is the method? Does it scale to e.g. aligning single-cell data, where normalization is often done simply be cell size?

- As mentioned before, a comparison with alternative methods, and a discussion on how to use the scaled data in downstream analysis would be of interest, but it is understandable if this is beyond the scope of this work. A particular question that may come up is: E.g. an ODE model will output values on a certain scale, which may be different from the normalized scale by the presented method. Would this necessitate the use of scaling factors when fitting the ODE model still?

Grammar

-------

- e.g. l. 8, 37: While this word also exists, you probably mean "Therefore" in multiple places.

- l. 37: "[,] hampering"

- l. 195: "concentrations[,] meaning"

- Table 1: comma in $Y_s = f^{-1}(Y, \\hat s)$

6. PLOS authors have the option to publish the peer review history of their article (what does this mean?). If published, this will include your full peer review and any attached files.

Reviewer #1: No

Reviewer #2: No

Reviewer #3: **Yes: **Yannik Schälte

---

## [Author Response · Author response to Decision Letter 0]

7 Jun 2022

Thank you for submitting your manuscript to PLOS ONE. After careful consideration, we feel that it

has merit but does not fully meet PLOS ONE’s publication criteria as it currently stands. Therefore, we

invite you to submit a revised version of the manuscript that addresses the points raised during the review

process.

The three reviewers agree that the manuscript would be a valuable addition to the literature, provided

that a number of aspects are clarified, and I concur with their assessment. In the review reports you will

find a list of suggestions to clarify a number of technical aspects of the proposed method. Notably, two of

them agree that it would be interesting to compare BlotIt with alternative approaches. As pointed out by

Reviewer 2, one possibility would be to perform a comparison using simulated datasets. If possible, such

a study would greatly enhance the contributions of the paper.

Answer: We thank the editorial board for the opportunity to submit a revised manuscript. A new chapter including 

a detailed method comparison with simulated data was added. We will elaborate the specifics below.

• A rebuttal letter that responds to each point raised by the academic editor and reviewer(s). You

should upload this letter as a separate file labeled ’Response to Reviewers’.

• A marked-up copy of your manuscript that highlights changes made to the original version. You

should upload this as a separate file labeled ’Revised Manuscript with Track Changes’.

• An unmarked version of your revised paper without tracked changes. You should upload this as a

separate file labeled ’Manuscript’.

Journal Requirements:

1. Please ensure that your manuscript meets PLOS ONE’s style requirements, including those for file

naming.

Answer: We checked and conformed to PLOS ONE’s style requirements and those for file naming. Please let us

know if a requirement is not met.

When you resubmit, please ensure that you provide the correct grant numbers for the awards you received

for your study in the ‘Funding Information’ section.

Answer: Thanks for checking this in detail! We indeed remarked that the grant number 031L004 was incomplete (should be 031L0048). 

We corrected this in the ‘Funding Information’ section. We hope that this is what you were referring to. Otherwise, please let us now.

3. PLOS ONE now requires that authors provide the original uncropped and unadjusted images underly-

ing all blot or gel results reported in a submission’s figures or Supporting Information files. This policy

and the journal’s other requirements for blot/gel reporting and figure preparation are described in detail

at https://journals.plos.org/plosone/s/figures#loc-blot-and-gel-reporting-requirements and

https://journals.plos.org/plosone/s/figures#loc-preparing-figures-from-image-files. When you submit your

revised manuscript, please ensure that your figures adhere fully to these guidelines and provide the origi-

nal underlying images for all blot or gel data reported in your submission. See the following link for in-

structions on providing the original image data: https://journals.plos.org/plosone/s/figures#loc-original-

images-for-blots-and-gels.

In your cover letter, please note whether your blot/gel image data are in Supporting Information or posted

at a public data repository, provide the repository URL if relevant, and provide specific details as to which

raw blot/gel images, if any, are not available. Email us at plosone@plos.org if you have any questions.

In this manuscript only simulated data and already published measurements from Kok et al. were used.

Answer: In this manuscript only simulated data and already published measurements from Kok et al. were used. 

There might have been a misunderstanding in Figure 1. We apologize for that and adjusted the figure description 

to clarify that simulated data was used.

Reviewers’ comments:

Reviewer #1: BlotIt - Optimal alignment of western blot and qPCR experiments

Summary

The authors present an automated procedure for the normalisation of relative data, such as data obtained

from Western blot or rt-qPCR. Relative data are usually not directly comparable across replicates, because

of different arbitrary units obtained, and so require normalisation. Using likelihood-based optimisation,

the proposed method is able to estimate how to scale each replicate, and at the same time provides an

estimate of mean and variance for the underlying data. The approach is flexible in various way, as it

allows different error models to be used, an also it is able to combine data from multiple blots as long as

there is one experiment shared across pairs of blots. The work presented is certainly worthy of publication,

although some clarifications and minor corrections are necessary before I can give my final approval.

Major

1. Missing discussion of the fact that Yij at extremes of the dynamic range of detection can have relatively

low signal to noise ratio or high residual error epsilon. While the error epsilon is initially introduced as

potentially different for each Yij, it is then simplified to have a variance proportional to the value of

the measurement (sigma ij = e rel ∗ yi/sj). While this can be considered an appropriate generic error

model, its drawbacks should also be highlighted, such as its underestimation of variance for relatively

low intensity blot values (which notoriously have a low signal to noise ratio). It would also be more

appropriate to adjust lines 80-83 page 3, to reflect the fact that this error model (Eq 2) efficacy depends

on the assumptions chosen to implement h and whether the data follows these assumptions, rather than

just describing it as a superior approach. The advantages and disadvantages of the chosen simplified

model should be highlighted. I guess the advantage here is the reduction of parameters to estimate (just

e rel), while the disadvantage is the loss of model flexibility. For example, it will ignore that low intensity

values might have a much higher relative variability (because of the low signal to noise ratio).

Answer: Thanks for commenting this point in detail! It is indeed important to adjust the error model according to

the error distribution of the data. We therefore added emphasis on the importance of the choice of error

model in the revised manuscript. For the specific case mentioned above the purely relative error model

σ_ij = e_rel · y_i /s_j still results in individual errors of Y_ij . We added a more detailed explanation why the

reduced error model is an appropriate choice for the discussed application. [revised manuscript lines 90-96]

2. Please define a set or range of values for the indexes i and j. For example there could be I conditions

and J blots with i in the range 1,. . . ,I and j in the range 1,. . . ,J. In principle, each equation should have

a definition for i and j range, like ‘for all i in (. . . ), j in (. . . )’. The simplest case would be when the

set of experiments i is the same for all J. However, the authors imply that different j can have different

sets of experiments, so different i? In this case perhaps it makes more sense to talk about indexes i that

belong to sets Ij (that is indexed by j) and that the intersections of sets Ij need to be at least pairwise not

empty (i.e. share at least one condition i). This might also affect the notation in equation 10.

Answer: A more formal introduction of the sets I and J was added. There is only one true value for each y_i ,

i ∈ I, but the reviewer is correct, in each experiment, e.g. on each gel (for western blotting) a different

subset of I can be measured. This is indeed the case for the simulation study where different scenarios

are elaborated. [revised manuscript lines 58-65]

3. How is the mean of the true values y dash (equation 10c) defined? This is particularly important if

there are different conditions i on different blots j (like in Figure 2), I guess this will be a mean across

all i regardless of what blot j they belong to?

Answer: The scaling of replicates can only be performed relative to each other. Thus, one additional constraint has

to be introduced to define the scaling parameters. The constraint fixes the mean over all measurements

e.g. to one. This is indeed arbitrary. The remaining degree of freedom requires the choice of a unit to

display the results, here a multiple of the means over all data.

4. In equation 7, page 4, the measurement error term is not shown. How do you expect the value of

Ysij be determined if the error is unknown? For example, arranging Eq 1, Ysij = (Yij+eij)*(s hat j).

Because the error is potentially different for each data point, how will the mean and sd of the aligned data

be affected? More explanation is given later, but here a clarification and showing the particular example

(f=Y/s) next the generic equation would help the flow. Perhaps, it would help to mention the ideas of

error propagation here, with examples and simplifying assumptions.

Answer: The error is not shown in equation (7) because we want to visualize the scaling of the measured values

Y from their own to a common scale via the model. We added a small look-ahead to the more detailed

explanation later. The aligned data is always on the common scale, which is why the propagation of the

corresponding errors is not discussed. [revised manuscript lines 130-134]

5. At lines 129 to 131 page 4, the authors assert that for dynamic models parameter estimation mean and

standard deviation is sometimes preferred, citing the case of low number of replicates. One could argue

the opposite, that if there are few replicates, then all data should be used for fitting, because the average

and sd alone may not properly represent a given data point. I suggest to remove or add more literature

or explanation to support the authors claim.

Answer: There seems to be a misunderstanding based on poor phrasing on our side. We concur with the reviewer:

In cases of small number of replicates the use of scaled data (i.e. data scaled to common scale) is not the

best input. In those cases the aligned data set (what we meant with ”means”) and the Fisher information

based confidence intervals (see Table 1) are better suited. We rewrote the mentioned passage to clarify

this. [revised manuscript lines 144-149]

6. The claim at lines 133 and 134 that the estimated errors are more reliable than the data spread obtained

from replicates, should be further explained. For example, this could be true only if the data agree with

the error models and there could be exceptions (see above for low signal to noise data points), and also it

depends on how the data spread of replicates is calculated, for example if the replicates are done all on

the same blot, then it might be more accurate, but if they are done on different blots, then the spread is

completely dependent on the normalisation applied and possibly the value of other data points.

Answer: This is a misunderstanding, the aligned data set consists of the estimated true values y . The eˆ describes

not the errors of the error model but the uncertainties of the model fit itself, and thus quantify the error

of fitted parameters yˆ. This is elaborated in the error calculation section, we clarified this in the text.

[revised manuscript lines 146-147]

7. Why there is no error term in equation 8? Also, in this case it would be useful to give a generic

idea of how the error can be inferred and follow Eq 8 with the concrete example f=Y/s, and simplifying

assumptions.

Answer: Equation (8) only shows the definition of the predicted data set. The error calculation (of all data sets)

is topic of the Error determination section. We added a remark pointing at this section and included a

more detailed example there. [revised manuscript lines 156-157]

8. On page 5 line 146, please reconsider and rephrase the claim that the optimal theta is obtained by

minimising the spread of the replicates, because the likelihood model is not just about trying to reduce the

spread (low sigma).

Answer: Yes, this was an oversimplification, thank you for the remark. [revised manuscript lines 160-162]

9. In the error determination section (page 6), equations 13 and 14 require a reference, please add.

Also, it would be nice to accompany these equations with concrete examples for the specific cases and

simplifying assumptions described (like f = Y /s). For example, sigma s , if I am not mistaken, simplifies

to sigma s ij = s j · σ ij .

Answer: References for equations 13 and 14 and the example for the present model have been added.

• Reference Bessel correction (revised manuscript line 187)

• Reference Gaussian error propagation (revised manuscript line 192)

• Reference Fisher Information (revised manuscript line 197)

10. Perhaps it would be of interest adding to the discussion some speculative yet useful cases. For

example, what happens when two blots share only one experiment, but in one blot j the measurement is

likely to have a high signal to noise ratio, while in the other blot j’ the same experiment has a very poor

signal to noise, perhaps because it is a low intensity value. Would the proposed model be able to propagate

the variance of the normalised data? Would there be enough data to constrain the model optimisation?

Answer: Thanks for bringing up the effect of different signal to noise ratios again! To elaborate this effect and

the influence of different numbers of overlap samples we included several scenarios in the performance

analysis described in the new section [Application to simulated data]

11. What other error models are available in blotIt besides e rel · value and e abs ?

Answer: BlotIt does not have a repository of pre-implemented (error) models. The user can freely define them.

Minor

12. Shouldn’t Equations 5 and 6 mirror the definitions in Equations 3 and 4? Equations 5 and 6 seem to

be a mix of the generic vector format Equations 1 and 2 and equations with specific indexes i and j such

as Eq 3 and 4. Please, choose one format for clarity, probably the format with the indexes i and j would

be more suitable for Eq 5 and 6 following the flow of the paper.

13. Line 114, page 4, what is the dimensionality of y hat, s hat and e hat? Probably this will be clearer

once the indexes are clarified (see above)

Answer: This is a good point, we use the index definition for equations (3-6) now. The meanings of i, and j are

defined in the beginning of the methods section, their dimensionality should be clearer now. [equations

3,4 changed vectors to indexes]

14. ‘Therefore’ might be more appropriate than ‘therefor’ at line 8 page 1, and also line 50 page 2, and

also in other places across the manuscript.

15. Line 118 page 4, form -> from

16. In equation 7 page 4, the variable Y s is undefined. I was initially confused by it because the text that

precedes it talks about the true values y. It might help to write a sentence giving a proper definition for

Y s , and accompany this equation with another equation exemplifying what Y s ij look like for the simplified

model f = Y /s and error e rel · Y ij /s j .

Answer: The grammar mistakes have been fixed and the text preceding equation (7) was clarified. The error is not

mentioned here, because we wanted to highlight the scaling of the measured data. The error propagation

is then covered later in the respective section. Also the explicit example for the western blot scaling

model can be found there.

Reviewer #2: This manuscript describes a normalization strategy for western blotting as well as many

other assay types that can put relative data from different experiments or replicates on the same quantita-

tive scale. This is needed often because experiment specific factors cause the scaling to not be comparable

between replicates or different experiments. A main claimed novelty is that the same condition need not

be contained in every experiment for normalization. Rather, each experiment needs to share at least one

point with one other experiment. I would have liked to have seen more application and discussion to data

sets that have such a feature, and showing where current methods fail. And perhaps some discussion of

how often this scenario is found in the literature and would be needed.

Answer: In principle, all measurements can be measured as perfect replicates (so that in N experiments, the exact

same biological setup is measured N times). In practice, biological conditions (treatments, targets, time

points etc.) vastly outnumber the capacity of one experiment. This gives rise to the need of extending

that limit of comparable biological samples by measuring only some samples as overlap to previous

experiments.

We added a simulation study to discuss the performance of blotIt compared to different other scaling

approaches. In that context we also introduced some usual scenarios, in which partial overlap between

experiments is necessary.

There are other points listed below that may be important to address:

1. What justification do the authors have for assuming normality in errors for western blots? How much

does that affect the conclusions of the paper? Could alternative error models be used and BlotIt still

functions well? How does that impact application to other data types?

Answer: We made some changes to better motivate the choice of the explicit error model used in the manuscript.

The validity of the error model must of cause be addressed for each application. Due to the flexible

nature of the utilized optimization approach, every error model can be used.

2. It is appreciated that the authors technique is claimed to work when data points are not shared between

all replicates and/or conditions. How common or rare this is for bench scientists performing replicates

or experiments was not discussed. What novelties or advantages does blotit have when data points are

shared among all replicates or experiments? Getting more clarity on that would help make the impact and

uptake of the paper clearer.

Answer: Thanks for pointing out that the need of having small overlap didn’t become clear yet! Indeed, it is a

very common scenario that bench scientists perform replicate measurements that are not shared between

all experiments. We added an explanation to the discussion and evaluated this point also in the new

section [Application to simulated data]. [revised manuscript lines 404-410]

3. How does the proposed approach differ from the scaling factor approach described here: 10.1038/msb.2009.4

? There is a general lack of comparison to other analysis methods which have been established and used

for quite a long time, as cited by the authors.

Answer: Wang et al (10.1038/msb.2009.4) used the strategy of optimal alignment. In our new section [Application

to simulated data] we analyse the performance of this and other methods in relation to blotIt and work

out similarities and differences.

4. The discussion of how to use the R code and format it seems more appropriate for detailed methods

section, not the results section.

Answer: This is a valid point which we also considered when deciding on the structure of the paper. However, as

we would like to emphasis the application of our method to various data types, we decided to describe it

in the results section.

5. In Figure 2, how do alternative methods for normalizing data compare to BlotIt?

6. Perhaps a simulated data study where data points are actually shared between all experiments, but are

hidden to see how BlotIt does, could be an effective analysis to demonstrate usefulness and also compare

to other normalization methods.

Answer: Thanks for bringing up this point! It motivated us to perform a method comparison that is now included

as new section [Application to simulated data]. However, the data sets described for the real world

scenarios in former Figure 2, now Figure 3, cannot be scaled with the other approaches as no biological

condition (dose and time point) is measured in all three experiments E1-E3. Therefore, we analyzed the

performance of different scaling methods based on simulated data sets where all methods are applicable.

7. How would one compare the common scale data as the output of BlotIt to model simulations that would

have a different scale (e.g. absolute concentrations)? Often comparison to and use with dynamical models

is cited in the paper as a main motivator, but discussion with respect to this is lacking.

Answer: Thanks for pointing out this unclarity! Indeed, it is recommended to include one scaling parameter for

the whole data set, instead of N experiment-specific scaling parameters necessary without normalization, 

in the model formulation. This enables the proper comparison of the common scale data and the

model simulations. We clarified this important point in the manuscript. [revised manuscript lines 140-143]

8. Therefor − > therefore

Answer: Thanks, we corrected it.

Reviewer #3: Review for the manuscript ”BlotIt - Optimal alignment of western blot and qPCR ex-

periments”:

The manuscript proposes a novel alignment method for relative data. Via optimization, it finds a version

of the data on a common scale. An implementation in R is provided. The manuscript is overall well

written and easy to read, while in some places it could be more specific. In my opinion, the new method

is interesting and will find usage, while it is maybe not a mayor conceptual breakthrough.

I have a few comments, which I think should be addressed in a revision:

Content

——

- it could be made clearer that, unlike e.g. the approach by Weber et al., the approach is essentially

model-free, i.e. only dependent on the data and a noise model, but not e.g. a post-hoc employed ODE

model.

Answer: This is a valid point! We included a paragraph in the discussion addressing this difference. [revised

manuscript lines 424-427]

- A comparison to e.g. the method by Weber, as well as the method by Degasperi, in a situation where it

is applicable, would be of interest, e.g. regarding predictions, efficiency and uncertainties. It is however

understandable if this is beyond the scope of this work.

Answer: Since the paper was originally addressed to a more application based audience we initially refrained from

including a dedicated performance comparison of different scaling methods, especially since our focus lied

on cases where the other methods are structurally not applicable. However, we see the added value in

such an analysis. So we performed a simulation study for cases where all methods are applicable and

evaluated the different performances. The setup of the simulation study can be found in the methods

section, while the outcome is analyzed in the results section. We however only included data-based nor-

malization approaches that are independent of any ODE model and added a short outlook concerning

the Weber method in the discussion. [revised manuscript lines 422-428 494]

- l. 35f: ”disadvantage of enlarging the parameter space drastically”: The approach by Weber (or also

later papers by Loos et al. ”Hierarchical optimization for the efficient parametrization of ODE models”

and Schmiester et al. ”Efficient parameterization of large-scale dynamic models based on relative mea-

surements”) appear to argue explicitly that the parameter space is effectively not enlarged by a hierarchical

formulation.

Answer: This is a good point we didn’t consider so far! As we would not regard hierarchical optimizing as the

standard approach – depending on the used tools and setup – the parameter space enlargement can still

be a problem. We addressed this in the discussion now. [revised manuscript lines 420-422]

- l. 36f: I did not understand ”estimates of the scaling parameters might be biased by the model equations[,]

hampering hypothesis testing and therefor[e] interpretation”.

Answer: When scaling parameters are optimized along with the model parameters of an ODE model (here: classi-

cally by estimating experiment-specific scaling factors), the model could resolve ”problems” as e.g. wrong

model assumptions by changing the scales. Equally a wrong scaling model might influence other model

parameters. This is difficult to disentangle.

- l. 82: ”the error model considers the variance information from all experimental data” as opposed to

”calculation of errors based on the spread of measurement values” does not get clear to me.

Answer: We rewrote this paragraph to be more specific, also concerning the choice of the error model. [revised

manuscript lines 83-100]

- l. 146: There appears to be a constant π missing in (10b), when deriving (10a) and (10b) from a normal

density, which affects the relative impact of both terms.

Answer: Thanks for pointing that out! We corrected it in the revised manuscript. However, the objective function 

is only changed by an additive constant that doesn’t influence the optimum or the curvature of the

optimization landscape:

l(p) = sum(wres(p)^2) + log(pi*sigma(p)^2) = sum(wres(p)^2) + log(sigma(p)^2) + log(pi) 

- l. 146: I think it could be clarified that ȳ in (10c) denotes the mean (over all data points?)?

Answer: ȳ represents the mean of the estimated true values as described in line 155 of the original draft and line

171 of the revised manuscript.

- l. 146: Where does the 10 −3 come from? This appears to be rather arbitrary and may affect how much

emphasis the method puts on normalization. How sensitive is the method with regard to it? Or can it be

chosen in a problem-specific manner? Or would there be an alternative formulation as an optimization

problem with explicit constraint ȳ = 1?

Answer: Based on our experience, optimization under the side constraint mean = 1 is often not very stable 

numerically. That’s why we introduced the constraint via 1 − ȳ in the objective function. As ȳ = 1 can be

always fulfilled exactly, the result is not sensitive to the penalization factor, here 10 −3 . Only the number

of iterations to the optimum can be changed by varying this factors. 10 −3 proved itself stable in practice.

- l. 146: Can (10c) be interpreted stochastically (as (9) claims to describe a probability density)?

Answer: No, it cannot. As described above, 10c is an auxiliary construct to incorporate the constrained optimiza-

tion.

- l. 158: ”This drastically improves numerical stability”: This is surely an accurate fact, yet a reference

may be good. Is the method applicable to negative data?

Answer: The method is also applicable to negative data. An example is given by the qPCR data set from our

application example. As this data is on log scale values can be negative as well. The alignment model

has to be adjusted accordingly as described.

- l. 167ff: References on the parameter/data ratio, the variance underestimation, and the Bessel correction

would be good.

Answer: For the data/parameter ratio, this is a conservative estimate based on our experience. A reference for

the bessel correction was added.

- l. 180: The confidence interval appears based on a local Taylor approximation given asymptotic normality

of the maximum likelihood estimate (with covariance matrix given by the inverse Fisher information

matrix). Conceptually, there should be alternative methods, e.g. based on Wilk’s theorem or sampling.

Maybe a contextualization would be good?

Answer: We also calculated the confidence intervals of the parameters using profile likelihood. This led to very

well defined parameter intervals, indicating that the calculation based on the Fisher information is in fact

a conservative approximation. We discussed the implementation of the profile likelihood method in blotit

but eventually decided to keep the conservative approximations as the additional computation time is

not justified for the standard user.

- l. 184: What do the authors mean by the FIM is ”represented by the Hessian”?

Answer: Since the FIM I(θ) is defined as the second derivative of the log-likelihood I(θ) := (d^2 l(θ))/(dθ^2)

we implement it by ”recycling” the Hessian H which is evaluated in the fitting process anyway, and since the parameters

for the last fitting step are the MLE, the evaluation of the Hessian of said step gives directly the Variances

of the MLE: Var(θ̂) = 1/I(θ̂) = 1/H(θ̂)

- implementation of the method: How is the optimization problem solved? Are gradients available? Does

the problem have multiple local optima?

Answer: The Implementation is via a trust-region optimizer, and the system is extremely well behaved. It usually

takes no more then 50 steps to converge, and a second optimum was never observed. This might be

because – contrary to dynamical modeling – the used scaling model is extremely close to reality.

- implementation of the method: How computationally expensive is the method? Does it scale to e.g.

aligning single-cell data, where normalization is often done simply be cell size?

Answer: It is quite expensive compared to the analytical approaches. We did not do an explicit run time analysis,

because we applied this method only to data sets of roughly equal size. We expect the run time to suffer

heavily for extreme complicated sets (very large data sets with little pairwise overlap for example). At

least this process is very well parallelizable.

- As mentioned before, a comparison with alternative methods, and a discussion on how to use the scaled

data in downstream analysis would be of interest, but it is understandable if this is beyond the scope of

this work. A particular question that may come up is: E.g. an ODE model will output values on a certain

scale, which may be different from the normalized scale by the presented method. Would this necessitate

the use of scaling factors when fitting the ODE model still?

Answer: The primary focus of this work was the scaling itself. This is why ODE based approaches are a little less

prominent in the discussion. To your specific question: Yes, scaling factors are still necessary. However,

just one scaling factor is needed for the whole scaled data set in contrast to individual scaling factors for

all data subset as it would be necessary without normalizing the data in advance. [revised manuscript

lines 140-143]

Grammar

——

- e.g. l. 8, 37: While this word also exists, you probably mean ”Therefore” in multiple places.

- l. 37: ”[,] hampering”

- l. 195: ”concentrations[,] meaning”

- Table 1: comma in Y s = f −1 (Y, ŝ)

Answer: Thanks for pointing out these comma and spelling mistakes! They are corrected in the revised manuscript.

---

## [Decision Letter · Decision Letter 1]

4 Jul 2022

BlotIt - Optimal alignment of Western blot and qPCR experiments

PONE-D-22-03759R1

Dear Dr. Kemmer,

We’re pleased to inform you that your manuscript has been judged scientifically suitable for publication and will be formally accepted for publication once it meets all outstanding technical requirements.

Kind regards,

Alejandro Fernández Villaverde, Ph.D.

Academic Editor

PLOS ONE

Additional Editor Comments (optional):

Reviewers' comments:

Reviewer's Responses to Questions

**Comments to the Author**

1. If the authors have adequately addressed your comments raised in a previous round of review and you feel that this manuscript is now acceptable for publication, you may indicate that here to bypass the “Comments to the Author” section, enter your conflict of interest statement in the “Confidential to Editor” section, and submit your "Accept" recommendation.

Reviewer #1: (No Response)

Reviewer #2: All comments have been addressed

Reviewer #3: All comments have been addressed

2. Is the manuscript technically sound, and do the data support the conclusions?

Reviewer #1: Yes

Reviewer #2: Yes

Reviewer #3: Yes

3. Has the statistical analysis been performed appropriately and rigorously? 

Reviewer #1: Yes

Reviewer #2: Yes

Reviewer #3: Yes

4. Have the authors made all data underlying the findings in their manuscript fully available?

Reviewer #1: Yes

Reviewer #2: Yes

Reviewer #3: Yes

5. Is the manuscript presented in an intelligible fashion and written in standard English?

Reviewer #1: Yes

Reviewer #2: Yes

Reviewer #3: Yes

6. Review Comments to the Author

Reviewer #1: I would like to thank the authors for the additional work and for replying to my previous questions, which I consider answered.

I just have a few additional point, just the first point is of major concern, and hopefully can be addressed easily.

1. I appreciate the additional simulation study, however I am concerned about the criteria chosen to evaluate and compare the methods. The authors write: “The performance of the individual methods was evaluated for each of the data realizations based on the spread of the scaled data”. Do the authors mean that the methods producing the normalised data with the narrowest standard deviation are preferrable? I think that one of the points of the mentioned Degasperi et al, was that data themselves have a spread, and that if we underestimate such spread this could also be problematic, for example making us believe that there is a difference between two conditions just because our assumptions have reduced the uncertainty of their mean value. So, I wonder whether the goal should be to prefer a method that produces a spread of the scaled data that is as close as possible to that of the simulated data.

2. The Methods section begins with the definition of the sets I and J, as well as measurements Yij. If I understand correctly, the key message here is that measurements Yij are comparable across the index i but not j. If so, this should be stated clearly, and some examples perhaps modified to avoid confusion. For example, the examples of biological effects include things that are comparable like different conditions and time points, but also things that are not comparable like different protein targets in a Western blot.

3. Line 99 of update text: “all experimental data, what allows for a reliable error”, change ‘what’ to ‘which’?

Reviewer #2: The authors have done a reasonable job of addressing the concerns raised in the review. I think the paper should be published and will be of use to biological data analysis ideas.

Reviewer #3: My comments to the first version have all been sufficeintly addressed, with some minor issues:

- l. 425 and l. 446: "bloIt", and in a few other places. I guess the method goes only by "blotIt"

- e.g. missing spaces and commas and shortforms like "didn't" in a few places in the newly added text

- 10^{-3} in (10c): I would recommend to include the answer to the question as part of the manuscript or supplement.

- same for the answer to how the optimization problem was solved

7. PLOS authors have the option to publish the peer review history of their article (what does this mean?). If published, this will include your full peer review and any attached files.

Reviewer #1: No

Reviewer #2: No

Reviewer #3: **Yes: **Yannik Schälte

---

## [Editor Report · Acceptance letter]

13 Jul 2022

PONE-D-22-03759R1 

BlotIt - Optimal alignment of Western blot and qPCR experiments 

Dear Dr. Kemmer:

I'm pleased to inform you that your manuscript has been deemed suitable for publication in PLOS ONE. Congratulations! Your manuscript is now with our production department. 

Kind regards, 

on behalf of

Dr. Alejandro Fernández Villaverde 

Academic Editor

PLOS ONE